# Recent Advances in Catalytic Synthesis of Benzosultams

**DOI:** 10.3390/molecules25194367

**Published:** 2020-09-23

**Authors:** Quan-Qing Zhao, Xiao-Qiang Hu

**Affiliations:** Key Laboratory of Catalysis and Energy Materials Chemistry of Ministry of Education & Hubei Key Laboratory of Catalysis and Materials Science, School of Chemistry and Materials Science, South-Central University for Nationalities, Wuhan 430074, China; zqqqhz@163.com

**Keywords:** sulfonamides, benzosultams, C–H activation, photoredox catalysis, cascade reaction

## Abstract

Benzosultams represent one category of multi-heteroatom heterocyclic scaffolds, which have been frequently found in pharmaceuticals, agricultural agents, and chiral catalysts. Given the diversely significant functions of these compounds in organic and medicinal chemistry, great efforts have been made to develop novel catalytic systems for the efficient construction of benzosultam motifs over the past decades. Herein, in this review, we mainly summarize the recent advances in the field of catalytic synthesis of benzosultams from 2017 to August of 2020, with an emphasis on the scopes and mechanisms of representative reactions.

## 1. Introduction

Sulfonamides constitute one of the most important antibiotics developed from non-natural sources [1,2]. Benzosultams, a subclass of bicyclic sulfonamides, are found in numerous biologically active compounds, pharmaceuticals, as well as agricultural agents [3,4,5,6,7]. They exhibit a wide spectrum of promising bioactivities, such as antibacterial, antidiabetic, anticancer, and vitro HPPD (4-hydroxyphenylpyruvate dioxygenase) inhibitory activities, which have been widely utilized as inhibitors and drugs [8,9,10,11,12]. Representative examples have been outlined in Scheme 1. Moreover, benzosultams can be employed as chiral auxiliaries in asymmetric synthesis [13]. They are also versatile building blocks for the assembly of a variety of diversely functionalized benzosultam derivatives and other heterocyclic systems. In this context, various transformations have been developed, such as dipolar cycloaddition [14,15], nucleophilic addition [16,17,18,19,20,21], transition metal-catalyzed C–H activation/annulation [22,23,24,25], Corey–Bakshi–Shibata reduction, and Aza-Darzens condensation [26,27] etc., which extensively expand their application range in the area of synthetic chemistry. Therefore, the development of convenient methods for the efficient preparation of benzosultams play important roles in drug discovery and asymmetric synthesis, which have attracted much attention from synthetic chemists.

Traditionally, benzosultam scaffolds can be readily constructed from some specialized substrates in one step, such as intramolecular C–H amination [28], dipolar cycloaddition [29], and intramolecular Michael addition [30]. In addition, multi-component or -step reactions have also been developed to prepare these important compounds [31,32,33,34,35,36,37,38,39]. In recent years, the exploration of catalytic methods enabled by transition metal or organic catalysis has been established as a robust tool for the synthesis of benzosultams in an efficient and region-divergent fashion. Taking advantage of the high and unique reactivity of free radicals, the radical-induced synthetic strategy has also been applied to construct benzosultams. Mondal et al. successively summarized the advances in sultam synthesis, biological activities, and synthetic applications (2000–2010, 2011–June 2017) [40,41]. In 2020, Grygorenko and colleagues comprehensively discussed the recent developments in the synthesis of saturated bicyclic sultams [42]. Taking into account the significance of benzosultam scaffolds, in this review, we provide an overview of catalytic synthesis of benzosultams from 2017 to August of 2020, with an emphasis on catalytic models, substrate scopes, and reaction mechanisms. Based on different catalytic modes, the discussion is mainly classified into two sections, including transition metal-catalyzed benzosultam synthesis and visible light photocatalytic benzosultam synthesis. Of note, visible light-induced catalytic synthesis, rarely covered in previous reviews, will be discussed here in detail. These reactions in which benzosultams are employed as starting materials for further transformations will not be discussed here.

## 2. Transition Metal-Catalyzed Benzosultam Synthesis

### 2.1. Synthesis of Benzosultams via C(sp^2^)–H Functionalization

Over the past several decades, transition metal (Fe, Co, Ni, Ru, Rh, Pd, Ir, and Pt)-catalyzed C–H activation and subsequent functionalization have been intensively investigated and established a powerful tool for the construction of structurally diverse heterocycles [43,44,45,46,47,48,49]. In 2010, Urabe and co-workers developed a palladium-catalyzed C–H activation/cyclization reaction of *Z*-bromoalkenes for the synthesis of benzosultams. It should be noted that the *Z*-bromoalkenes can be easily synthesized from the nucleophilic addition of *N*-alkyl-*p*-toluenesulfonamides to bromoacetylenes [50]. Later, Mondal and co-workers further applied this strategy into an intramolecular cyclization of aryl bromides for the regioselective formation of uracil-, coumarin-, and quinolone-fused benzosultams [51]. Using three equivalents of AgOAc as the efficient oxidant, in 2014, Laha et al. demonstrated a Pd-promoted intramolecular oxidative coupling reaction of sulfonanilides, which offers a facile access to a range of biaryl sultams [52]. This protocol could be applied to the preparation of seven-membered biaryl sultams and biaryl sultones. The same strategy has been further used in the intramolecular oxidative coupling of *N*-arylsulfonyl heterocycles, such as *N*-arylsulfonyl indoles and pyrroles, producing heterobiaryl sultams in generally good yields [53], which may provide new opportunities in drug discovery. In addition, in the presence of amines, NaOEt, or Grignard reagents as nucleophiles, the ring opening of indole-fused benzosultams has been achieved for the preparation of heterobiaryls.

Inspired by these works, in 2017, Laha et al. systematically investigated the intramolecular oxidative coupling of diversely substituted *N*-arylsulfonyl pyrroles, delivering a variety of pyrrole-fused benzosultams **2** with good region selectivity (Scheme 2a) [54]. In this protocol, *N*-arylsulfonyl pyrrole **1** bearing a sensitive 3-bromo group was tolerated, albeit with the depression of reaction efficiency. Compared to indole-fused benzosultams, the ring opening of pyrrole-fused benzosultams turned out to be difficult and the introduction of an electron-withdrawing group at the C-2 position can significantly improve the reaction efficiency. To further broaden the synthetic utility of this strategy, the late-stage diversification of pyrrole-fused benzosultams has been investigated (Scheme 2b). Suzuki coupling reaction of pyrrole-fused benzosultam **2e** with arylborates proceeded smoothly to afford the fluorene tethered compound **4** in a 70% yield. The absorption and photoluminescence spectra of **4** displayed λ_max_ at 321 nm and λ_emission_ at 456 nm in dichloromethane, respectively, which has tremendous application potential in organic light-emitting devices (OLEDs) [55].

Later, Wang et al. discovered a Pd(II)-catalyzed cyclization of bioactive peptidosulfonamides through peptide-guided C–H activation (Scheme 3) [56]. The peptides acted as internal directing groups in this reaction, which enabled the selective cyclization of benzosulfonamides **5** and macrocyclization of peptidosulfonamides **8**. Under the reaction conditions, a variety of benzosultam-peptidomimetics **7** and peptidosulfonamide macrocycles **9** could be furnished in moderate to good yields. The potential utility of this methodology was demonstrated by the successful synthesis of a fluorescent-labeled cyclic RGD (Arg-Gly-Asp) peptide, which exhibits strong binding affinity towards integrins [57].

Mechanistic studies indicated the involvement of a reactive dipeptide–Pd(II) complex **5**-**E** in this transformation. As outlined in Scheme 4, the reaction starts with the formation of Pd(II)-complex **5**-**A** via Pd(II)-catalyzed C–H activation directed by the *N*-sulfonated peptide. Then, the coordination of intermediate **5**-**A** with olefin **6** generates complex **5**-**B**, which undergoes a 1, 2-migratory insertion process, followed by a β-hydride elimination to provide olefination intermediate **5**-**D**. In the presence of Pd(II)-catalyst, the resulting intermediate **5**-**D** performs another C–H activation, providing the key intermediate **5**-**E**. The structure of **5**-**E** was unambiguously confirmed by X-ray analysis. Finally, reductive elimination of **5**-**E** produces the desired product along with the formation of Pd(0)-catalyst. The reoxidation of Pd(0)-catalyst by Ag(I) or Cu(II) species could regenerate the active X_2_Pd(II)L_n_ catalyst for the next catalytic cycle.

Rhodium-catalyzed C–H activation/cyclization provides an alternative method for the assembly of heterocycles [43]. In 2012, Cramer and co-workers reported the first example of Rh(III)-catalyzed C–H activation of acylated sulfonamides and subsequent coupling with alkynes, delivering a broad range of six-membered cyclic benzosultams in generally good yields [58]. Then, Li and Wang et al. extended this strategy to the oxidative coupling of *N*-tosylacetamide with various acrylate esters for the preparation of five-membered cyclic benzosultams [59]. Using RuCl_3_ as an efficient catalyst, in 2018, You and colleagues disclosed a direct oxidative C–H/C–H cross-coupling of (hetero)aromatic sulfonamides with a series of (hetero)arenes for the synthesis of *ortho*-sulfonamido bi(hetero)aryls [60]. The obtained bi(hetero)aryl sulfonamides could be conveniently transferred into benzosultam derivatives in two steps.

Encouraged by these works, Chen and co-workers achieved a rhodium-catalyzed *ortho*-alkynylation or alkynylation/cyclization of arylsulfonamides, synthesizing an array of aryl sulfonamides and benzosultams (Scheme 5) [61]. The substituent in bromoalkyne has a significant effect on this reaction. When (triisopropylsilyl)ethynyl bromide was used as the substrate, the corresponding *ortho*-(1-alkynyl) benzenesulfonamides could be obtained. Interestingly, using triethyl (TES) or trimethyl (TMS) protected bromoacetylenes as coupling regents, a cascade *ortho*-alkynylation/cyclization of arylsulfonamides could occur to give six-membered benzosultams (Scheme 5a). One impressive feature of this reaction is the broad substrate scope. Aryl sulfonamides bearing an electron-donating or -withdrawing group at different positions were well-tolerated in this reaction. It is noteworthy that the protecting groups could be readily removed from benzosultam products in the presence of *tetra*-*n*-butylammonium fluoride (TBAF), enlarging the application range of these reactions (Scheme 5b). In addition, the gram-scale reaction was also successful.

Mechanistic studies revealed the Rh-catalyzed *ortho*-C–H cleavage was the rate determining step in this reaction. The coordination of Ru(III)-catalyst with arylsulfonamide affords a five-membered rhodacycle species **10**-**B** via *ortho*-C–H activation (Scheme 6). Then, the dissociative ligand exchange of **10**-**B** with **11** results in **10**-**C**, which undergoes an alkyne insertion process to provide **10**-**D**. The subsequent elimination of Br-atom of intermediate **10-E** by AgOAc produces complex **10**-**F**, followed by ligand exchange with substrate **10** to deliver the product and regenerate rhodacycle complex **10**-**B**.

Despite the impressive achievements in the synthesis of intriguing five- or six-membered sulfonamide-fused skeletons, the development of efficient methods for the assembly of larger macrocyclic ring frameworks remain largely unexplored and a challenging task for synthetic chemists. In 2020, Zhou and Yi et al. discovered an Rh-catalyzed intramolecular dehydrogenative annulation of 2-alkenylanilides to give various eight-membered benzosultams with an excellent chemoselectivity through PivOH-assisted C−H activation and the H-transfer process (Scheme 7) [62]. It was found that the rational combination of [Cp*^Cy^RhCl_2_]_2_ catalyst and KOPiv can efficiently promote this transformation. This catalytic system was effective for intramolecular dehydrogenative cyclization of *N*-(2-vinylphenyl)benzamides, furnishing the corresponding eight-membered lactams in moderate yields. Mechanistic investigations by density functional theory (DFT) calculations and control experiments suggested the Rh(III)−hydride species was the active catalyst in this reaction. In addition, H_2_ proved to be the major byproduct by PdCl_2_ testing paper. Alternatively, cobalt-catalyzed aryl C(sp^2^)−H activation/annulation have also been developed for the construction of benzosultam scaffolds. For instance, Sundararaju and Volla et al. independently developed Co-catalyzed C−H activation/N−H cyclization of sulfonamides with the use of alkynes or allenes as coupling partners, yielding a variety of biologically interesting benzosultam motifs [63,64]. These works have been covered in a previous review, which were not discussed in detail here [41].

Benzo-γ-sultam represents a quite unusual type of benzosultams, which is a key structural motif in ORL1-receptor (opioid receptor-like 1) antagonists, which have been used for the treatment of pain and CNS (central nervous system) disorders. In addition, benzo-γ-sultams can also be regarded as versatile building blocks for the preparation of heterocycles through the generation of reactive *aza*-*ortho*-xylylene species by thermal extrusion of SO_2_ [65]. Early in 1963, Bunnett and co-workers firstly synthesized benzo-γ-sultams via an intramolecular ring closure of *N*-2′-chloroaryl alkanesulfonamides under strongly basic conditions [66]. Since then, great efforts have been devoted into this interesting field [67,68]. In 2014, Xu et al. developed an Rh-catalyzed intramolecular C(sp^2^)–H insertion of *N*,*N*-diaryl diazosulfonamides, providing *N*-aryl-substituted benzo-γ-sultams in moderate to good yields (Scheme 8a) [69]. Outstanding features of this protocol include high efficiency and low catalyst loading (0.5 mol%). However, this rhodium-mediated catalytic system turned out to be ineffective for *N*-alkyl-*N*-aryl diazosulfonamide substrates. Afterwards, Xu and Yang et al. addressed this problem with the use of an inexpensive copper-catalyst in reflux toluene (Scheme 8b) [70].

### 2.2. Synthesis of Benzosultams via C(sp^3^)–H Amination

The selective functionalization of C(sp^3^)–H bonds is one of the most attractive methods for molecular construction in modern synthesis. However, the activation of inert C(sp^3^)–H bonds remains a fundamental challenge for the synthetic community due to their robust bond energy [49,71]. In 2007, Zhang’s group firstly developed an efficient Co-based catalytic system for the intramolecular C(sp^3^)–H amination with arylsulfonyl azides, resulting in a range of highly valuable benzosultams under mild conditions [72]. Primary, secondary, and tertiary C–H bonds were compatible with this catalytic system. In 2010, Zhang et al. further applied this concept to the amination of sulfamoyl azides for the high-yielding preparation of cyclic sulfamides [73].

Very recently, the asymmetric variants of intramolecular C(sp^3^)–H amination have been achieved by the same group with the use of Co(II)-catalysts of D_2_-symmetric amidoporphyrins, producing five-membered chiral cyclic benzosultams in good yields with high enantioselectivities (Scheme 9) [74]. Importantly, this reaction can be scaled up to 10 mmol to give the desired product without affecting reaction efficiency. In addition, this methodology was successfully applied in the concise synthesis of a chiral-fused-tricyclic sulfonamide target, which exhibits a broad range of enzyme inhibitory properties. These applications nicely demonstrated the practicality and synthetic utility of this 1,5-C–H amination reaction. A series of mechanistic studies have been conducted to investigate the unique metalloradical mechanism of this reaction. The results of EPR experiments demonstrated the intermediacy of a α-Co(III)-aminyl radical in this transformation. The high value of the intramolecular kinetic isotopic effect (KIE) suggested that the cleavage of C(sp^3^)–H bonds was realized through H-atom abstraction by α-Co(III)-aminyl radical species. On the basis of these results, a plausible mechanism is proposed in Scheme 9b. Initially, cobalt catalyst reacts with sulfonyl azides **21** to generate α-Co(III)-aminyl radical **21**-**A**. Then, the species **21**-**A** undergoes a 1,5-H atom abstraction to afford **ε**-Co(III)-alkyl radical **21**-**B**, followed by a radical substitution step to give the final product.

Cytochrome P450 enzymes, produced from thermophilic organisms, represent a huge family of oxidative hemoproteins, which have been often applied in oxygen-atom transfer reactions [75,76,77]. Arnold and Fasan et al. independently found that cytochrome P450 enzymes can be used as efficient nitrene transfer catalysts to promote intramolecular C(sp^3^)–H amination with sulfonylazides for the synthesis of benzosultams with good enantioselectivities [78,79,80,81,82]. However, among these reactions, the yields of amination products are relatively low due to the competing reduction of sulfonyl azides to sulfonamides. Recently, Hartwig et al. utilized artificial cytochrome P450 enzymes, containing an iridium porphyrin cofactor (Ir(Me)-PIX), as an efficient catalyst to realize the insertion of nitrene into the C(sp^3^)–H bond with excellent chemoselectivity over the reduction of sulfonyl azides (Scheme 10) [83]. Of note, the change of the active site in P450 enzyme from iron to non-native metal iridium creates a highly active catalyst for chemoselective C–H amination. Under mild conditions, the desired benzosultams can be obtained in up to a 98% yield and 294 TON (turnover number).

More recently, Schomaker and co-workers developed a silver-catalyzed nitrene transfer for the amination of C(sp^3^)–H bonds (Scheme 11) [84], providing a series of five- and six-membered benzosultams. The regioselective differentiation of vicinal methylene C(sp^3^)–H bonds could be achieved via electronic and steric control. The amination of homobenzylic methylene C–H bonds proved to be more preferential than their benzylic methylene neighbors. For alkyl-substituted benzenesulphonamides, six-membered benzosultams can be obtained as the main products. Moreover, when γ-branched alkyl substituents were used as the substrates, ligand-controlled tunable regioselectivity was observed for the synthesis of five- or six-membered benzosultams. It is noted that this reaction should be protected from light with Al foil and the yield of products can be significantly improved in the absence of light.

### 2.3. Synthesis of Benzosultams via Hydroamination of Alkyne

Transition metal-mediated hydroamination of alkyne provides a powerful platform for the formation of C–N bonds, which has found widespread applications in the synthesis of various biologically important heterocycles [85,86]. In 2015, Mondal and co-workers developed an elegant Pd/Cu-catalyzed one-pot Sonogashira coupling/hydroamination cascade of 2-bromobenzenesulfonamides with terminal alkynes, providing an alternative access to benzosultams [87]. In addition, Zhu et al. finished the synthesis of a variety of indole-fused benzosultams via Pd(II)-catalyzed diamination of alkynes [88].

In 2018, Mondal and co-workers further extended their one-pot process towards the construction of indole-fused seven-membered benzosultams from easily available starting materials (Scheme 12) [89]. The synergistic combination of Pd(PPh_3_)_2_Cl_2_ and CuI catalysts in DMF was found to be crucial for high reaction conversion. Under the optimal conditions, a variety of indole-fused benzosultams were obtained in good yields. In addition, the role of the *N*-protecting group in *o*-iodoanilines has been investigated. It should be noted that *N*-mesylated *o*-iodoanilines produced the free indolyl nitrogen-containing benzosultams, while N-unsubstituted-*o*-anilines failed to give the cylization products. The plausible mechanism is outlined in Scheme 12. Firstly, a Pd/Cu-catalyzed Sonogashira coupling reaction occurs between *o*-iodoaniline **31** and propargylsulfonamides **32** to yield intermediate **32**-**A**. Then, the Cu(I)-catalyst may coordinate with alkynyl and sulfone groups to produce complex **32**-**B**, which undergoes an intramolecular nucleophilic attack to give intermediate **32**-**C**. The resultant intermediate further proceeded an oxidative addition, intramolecular arylation, and reductive elimination sequence to generate intermediate **32**-**F**, followed by a deprotonation process to give the final product.

With the assistance of a catalytic amount of silver catalyst (AgSbF_6_), Reddy and co-workers firstly synthesized a series of fused benzo-δ-sultams from alkynols and aldehydes via silver-promoted hydroamination/Prins-type cyclization strategy (Scheme 13) [90]. The selection of a catalyst is critical for the success of this reaction and AgSbF_6_ proved to be the most preferable catalyst. Other Lewis acids (In(OTf)_3_, FeCl_3_, TMSOTf, and Sc(OTf)_3_) or Brønsted acids (*p*-TSA and TFA) failed to promote this transformation. A range of aliphatic and (hetero)aromatic aldehydes were suitable for this reaction, giving the expected compounds in moderate to good yields. Generally, (hetero)aromatic aldehydes delivered the products in higher yields than aliphatic aldehydes. Remarkably, the acid-sensitive cinnamaldehyde was well-tolerated in this reaction to give the corresponding product in excellent yield. This reaction provides a facile and straightforward access to fused benzo-δ-sultams. The reaction mechanism initially starts with the coordination of Ag(I)-catalyst with the alkyne group, producing an Ag–π complex **34**-**A** (Scheme 13b). An intramolecular 6-*endo*-dig cyclization of intermediate **34**-**A** occurs to form the intermediate **34**-**B**. A condensation reaction between **34**-**B** and aldehyde affords the reactive *oxo*–carbenium ion species **34**-**C**, followed by a Prins-type cyclization to give the final products **36**.

Very recently, Blanc and co-workers developed a convenient approach to benzosultams from *N*-(2-alkynyl)-phenylsulfonyl azetidines and nucleophilic alcohols through a gold(I)-mediated cyclization/nucleophilic substitution process (Scheme 14) [91]. A variety of alkyl alcohols were compatible with this catalytic system. However, phenol was unable to proceed this reaction due to its weak nucleophilicity. Notably, indole, a C-based nucleophile, could also provide the corresponding product **38e** at an elevated temperature. Other strong nucleophiles, such as amines and thiols, were unsuccessful substrates for this reaction, which may cause the deactivation of gold(I)-catalyst. When terminal alkyne was subjected into this reaction at 70 °C, the unexpected product **39** was observed.

Mechanistic investigations, including blank and isotopic labeling experiments, revealed that the spiroammonium intermediate **37**-**B** is a plausible intermediate in this reaction. Moreover, the intermediate **37**-**B** can be easily trapped by *N*-iodosuccinimide (NIS) to deliver 4-iodobenzosultams, which subsequently participated in cross-coupling reactions for the facile construction of highly functionalized benzosultams. As described in Scheme 14b, the activation of acetylenic moiety by gold(I)-catalyst generates species **37**-**A**, which reacted with sulfamide fragment to give spiroammonium **37**-**B** via a nucleophilic addition process. Then, the nucleophilic attack of **37**-**B** by protic nucleophiles results in the intermediate **37**-**C**. The protonation of **37**-**C** delivers the final product and regenerate gold(I)-catalyst.

## 3. Visible Light Photocatalytic Synthesis of Benzosultams

In recent years, visible light-photoredox catalysis has emerged as an ideal tool for the generation of radical species. Taking advantage of the unique and high reactivity of radical intermediates, visible light-induced catalytic synthesis provides new synthetic opportunities to reaction invention and complex molecule construction, which offers good solutions to unlock previously inaccessible reactions [92,93,94,95,96,97,98,99,100,101,102,103,104,105,106,107,108].

Under mild visible light-photocatalytic conditions, in 2016, Wu and Kuang et al. achieved a facile synthesis of benzosultams from 2-ethynylbenzenesulfonamides and Togni’s reagent (Scheme 15) [109]. Ir(ppy)_3_ was demonstrated as the best choice of photocatalyst, while other commonly used catalysts, such as Ru(bpy)_3_(PF_6_)_2_ or eosin Y, were ineffective. This reaction displayed a broad substrate scope. For example, 2-ethynylbenzenesulfonamides bearing electron-donating or -withdrawing groups at aromatic ring were smoothly transferred into benzosultam products in good yields with relatively low *E*/*Z* selectivity. Importantly, increasing the steric hindrance of the protecting groups on the nitrogen atom could largely improve the *E/Z* selectivity. When cyclohexyl, *tert*-butyl, and *sec*-butyl groups were used, (*E*)-benzosultams could be obtained as the major products.

As depicted in Scheme 16, upon the irradiation of visible light, the ground state of photosensitizer Ir(ppy)_3_ can be transferred into its excited state *[Ir(ppy)_3_], which undergoes an oxidative quenching by Togni’s reagent **41** to generate radical anion intermediate **41**-**A.** Then, the C–I bond homolytic cleavage of intermediate **41**-**A** forms a trifluoromethyl radical. The rapid radical addition of trifluoromethyl radical to 2-ethynylbenzenesulfonamide **40** would offer the alkenyl radical **40**-**A**, followed by a SET oxidation pathway to give alkenyl cation intermediate **40**-**B** along with the regeneration of the ground state of Ir(ppy)_3_, thus completing the photocatalytic cycle. Finally, under basic conditions, there is the intramolecular nucleophilic addition of intermediate **40**-**B** to produce the benzosultam product.

Shortly afterward, using diazonium salts as a radical source in place of Togni’s reagent, Alcaide and Almendros et al. finished the synthesis of 3,4-diaryl-benzosultams with satisfactory yields by cooperative gold and photoredox catalysis (Scheme 17) [110]. This dual catalytic system can be applied for the divergent construction of other heterocyclic cores, such as benzothiophenes, isocoumarins and 3H-indoles, which highlighted the utility of this methodology.

The synthesis of benzosultams under metal-free conditions is undoubtedly appealing in organic and medicinal chemistry. Very recently, Zeng and co-workers developed a photo/disulfide-induced carbooxygenation of *N*-arylsulfonylamido alkynes by employing Eosin Y as the organic photocatalyst and molecular O_2_ as a green oxygen source, offering an efficient way to access benzosultams (Scheme 17) [111]. A range of functional groups, such as trifluoromethyl, chloro, and bromo, were compatible with the reactions to give variously substituted benzosultams in acceptable yields. The key to its success was the addition of PhSSPh for the in situ generation a thiyl radical. The authors found that the reaction could occur to give a 48% yield of product in the absence of Eosin Y and no reaction was observed without PhSSPh. As outlined in Scheme 18, the reaction starts with the formation of a thiyl radical through visible light-induced homolysis of PhSSPh. Meanwhile, upon the irradiation of 30 W blue LEDs, the reductive quenching of photoexcited Eosin Y* (EY*) by PhSSPh provides another pathway to generate the thiyl radical. The subsequent radical addition of thiyl radical to the alkyne group results in a C–Centered radical **46**-**A**, followed by a 5-*exo*-cylization, SET oxidation, 1,2-migration, and aromatization cascade to give intermediate **46-F**. In addition, radical **46**-**A** can also undergo a sequential 6-*endo*-cylization, SET oxidation, and aromatization process to deliver the same radical intermediate **46**-**F**. Then, thiyl radical abstracts a H-atom from **46**-**F** to form radical **46**-**G**, followed by an isomerization process to generate **46**-**H**. The resultant **46**-**H** further reacts with molecular O_2_ to give the final product **48**.

In addition, Xiao and Chen et al. developed a dual catalytic system for the efficient preparation of dihydropyrazole-fused benzosultams from β, γ-unsaturated hydrazones by the merging of visible light photoredox catalysis and cobalt catalysis (Scheme 19) [112]. In the absence of cobalt catalyst, this visible light-induced N-radical reaction proceeded smoothly, delivering the unsaturated compound **51**, which is unstable and can be slowly oxidized to the aromatic product **50** under air conditions. The addition of a suitable cobalt catalyst plays an important role to facilitate this aromatization process by the generation of H_2_ as the sole byproduct, avoiding the use of other external oxidants. A series of aromatic or aliphatic group-substituted β, γ-unsaturated hydrazones participated in this reaction smoothly, which demonstrated the generality of this cascade reaction.

Radical-trapping experiments by TEMPO indicated the involvement of a C-based radical, which was believed to be generated from the intramolecular addition of an N-centered radical to C=C bonds. The fluorescence quenching studies disclosed that the photoexcited *[Ru(bpy)_3_]^2+^ was efficiently quenched by the nitrogen anion generated from the deprotonation of β, γ-unsaturated hydrazones. Taken together, a visible light-induced N-radical 5-*exo* cyclization, addition, and aromatization pathway is proposed. As depicted in Scheme 20, the initial deprotonation of hydrazone generates nitrogen anion intermediate **49**-**A** under the basic conditions. Meanwhile, upon the irradiation of 3 W blue LEDs, the photoexcitation of the ground state of photocatalyst [Ru(bpy)_3_]^2+^ generates *Ru^2+^ species, which is quickly quenched by N-anion **49**-**A** to give N-centered radical **49**-**B**, together with the formation of reduced Ru^1+^ catalyst. Then, radical **49**-**B** undergoes an intramolecular cyclization and addition cascade to give radical species **49**-**D**. The Co(III)-catalyst can regenerate the ground state of [Ru(bpy)_3_]^2+^ through a SET oxidation, closing the photocatalytic cycle along with generating Co(II)-catalyst. The oxidation of intermediate **49**-**D** by Co(II)-catalyst results in cationic intermediate **49**-**E** and Co(I)-catalyst. Under the basic conditions, the deprotonation of the resultant **49**-**E** produces the final product. The combination of Co(I)-catalyst with a proton gives rise to Co(III)–H intermediate, which may couple with another proton to release H_2_ and regeneration of the Co(III) catalyst, thus completing the cobalt catalytic cycle. The reduction of radical **49**-**D** by [Ru(bpy)_3_]^1+^ delivers intermediate **49-F**, followed by a protonation process to give the unsaturated compound **51**.

## 4. Conclusions

Benzosultams, a subclass of bicyclic sulfonamides, are privileged skeletons in various biologically active compounds and chiral catalysts. The development of efficient methods for their synthesis plays a vital role in new reaction invention and drug discovery. The purpose of this review was to summarize the recent achievements in the field of catalytic synthesis of structurally diverse benzosultams from 2017 to August 2020, with an emphasis on catalytic models, substrate scopes, and reaction mechanisms. Accordingly, transition metal-catalyzed strategies for benzosultam synthesis, including C(sp^2^)–H functionalization, C(sp^3^)–H amination, and alkyne hydroamination were highlighted in Section 2. Visible light-induced catalytic strategies, which are rarely covered in previous reviews, were discussed in detail in Section 3. These reactions provide convenient and alternative platforms for the construction of benzosultams.

Despite these impressive advances, there are still some challenges in this field as follows: (1) Catalytic asymmetric synthesis of benzosultams; and (2) large-scale synthesis. We believe that the further exploration of novel dual catalytic systems by rational combination of transition metal or organocatalysis with photoredox catalysis may provide good solutions to the mentioned problems. The continuous flow processing can greatly improve reaction efficiency, thus facilitating large-scale synthesis. We hope this review will inspire more interest in the development of efficient catalytic strategies for benzosultam synthesis.

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
