# Peer review of "Recent Advances in Catalytic Synthesis of Benzosultams"

_molecules, 2020, doi:10.3390/molecules25194367_

Round 1
Reviewer 1 Report
The synthesis of benzosultams is still an important topic in organic synthesis because of their usefulness of pharmaceuticals, agricultural compounds, and chiral auxiliaries. This review is described mainly in two parts: (1) transition metal-catalyzed strategy including C(sp2)–H functionalization, C(sp3)–H amination and alkyne hydroamination and (2) visible light-induced catalytic strategy for the synthesis of benzosultams.
This review is suitable for broad readers in the field of organic, pharmaceutical and agricultural chemistry.
Please check the following points.
Line 82. “---- afford the fluorine tethered compound 4 in 70% yield.” Compound 4 does not have fluorine atoms. Is Scheme 2b correct?
Some abbreviations are used without the technical terms in full.
The information is needed in the text because review articles are for broad readers.
Line 93. What does “RGD” stand for?
Lines 160 and 212. What do “ORL1”, “CNS”, and “TON” stand for?
Author Response
Reviewer 1: The synthesis of benzosultams is still an important topic in organic synthesis because of their usefulness of pharmaceuticals, agricultural compounds, and chiral auxiliaries. This review is described mainly in two parts: (1) transition metal-catalyzed strategy including C(sp2)–H functionalization, C(sp3)–H amination and alkyne hydroamination and (2) visible light-induced catalytic strategy for the synthesis of benzosultams. This review is suitable for broad readers in the field of organic, pharmaceutical and agricultural chemistry. Please check the following points:
(i) Line 82. “---- afford the fluorine tethered compound 4 in 70% yield.” Compound 4 does not have fluorine atoms. Is Scheme 2b correct?
(ii) Some abbreviations are used without the technical terms in full. The information is needed in the text because review articles are for broad readers. Line 93. What does “RGD” stand for? Lines 160 and 212. What do “ORL1”, “CNS”, and “TON” stand for? My colleagues and I thank this referee very much for his/her favorable comments and many helpful suggestions! I have carefully revised the manuscript according to these suggestions.
Q1. Page 3, Line 82. “---- afford the fluorine tethered compound 4 in 70% yield.” Compound 4 does not have fluorine atoms. Is Scheme 2b correct? Response: This is a spelling mistake. We changed the “fluorine” to “fluorene”.
Q2. Page 3, Line 93. What does “RGD” stand for? Response: RGD stands for Arg-Gly-Asp containing peptides. According this important suggestion, we added the information in revised manuscript.
Q3. Page 6, Lines 160. What do “ORL1” and “CNS” stand for? Response: ORL1-receptor stands for opioid receptor-like 1-receptor and CNS stands for central nervous system. We added the corresponding technical terms in the main text.
Q4. Page 8, Lines 212. What does “TON” stand for? Response: TON stands for turnover number. According this suggestion, we added this information in revised manuscript.
Reviewer 2 Report
Manuscript ID: Molecules-938089
Manuscript title: “Recent advances in catalytic synthesis of bezosultams”
By Quan-Qing Zhao and Xiao-Qiang Hu
Dear Editor,
The paper “Recent advances in catalytic synthesis of benzosultams” involves the review of recent studies of catalytic synthesis of benzosultams.
The paper is very well written focusing two principal ways: 1. the benzosultams synthesis by metal transition-catalyse by across C(sp2)-H functionalization, C(sp3)-H amination and hydroamination of alkynes, and 2. Synthesis of benzosultams via visible-light catalysis.
In my opinion the paper should be accepted.
Author Response
Reviewer 2: The paper “Recent advances in catalytic synthesis of benzosultams” involves the review of recent studies of catalytic synthesis of benzosultams. The paper is very well written focusing two principal ways: 1. the benzosultams synthesis by metal transition-catalyse by across C(sp2)-H functionalization, C(sp3)-H amination and hydroamination of alkynes, and 2. Synthesis of benzosultams via visible-light catalysis. In my opinion the paper should be accepted.
My colleagues and I thank this referee very much for his/her favorable comments
Reviewer 3 Report
This paper is the very rare case of manuscripts, which are interesting, well written and organized. After some editorial control of the English, it sholuld be published as it is.
Author Response
Reviewer 3: This paper is the very rare case of manuscripts, which are interesting, well written and organized. After some editorial control of the English, it sholuld be published as it is.
My colleagues and I thank this referee very much for his/her favorable comments.
Response: Thank you very much for your kind suggestion. Accordingly, we modified the English in the abstract part.